# Backward-Facing Analysis for the Preliminary Estimation of the Vehicle Fuel Consumption

Stefan Tabacu *  and Dragos Popa *

Department of Road Vehicles and Transports, University of Pitesti, 1 Targu Din Vale, 110040 Pitesti, Romania
* Correspondence: stefan.tabacu@upit.ro (S.T.); dragos.popa@upit.ro (D.P.)

**Abstract:** In this paper, a methodology for the estimation of fuel consumption using backward-facing analysis is presented. The method for the determination of fuel consumption was based on the evaluation of the total work required to drive the vehicle along a specific drive cycle. At the same time, the potential fuel economy was estimated using the energy that can be harvested from the wheel. The results obtained using this model were compared with complex MATLAB/Simulink models developed using the forward-facing strategy. The MATLAB Simulink model details conventional and hybrid vehicle models capable of estimating fuel consumption. The components of the longitudinal forces opposing the vehicle during driving were investigated and analyzed using the results of the analytical and simulation models. The procedure can be successfully applied to provide a quick estimate of the fuel consumption performance before detailing complex simulation models.

**Keywords:** backward-facing analysis; fuel consumption; longitudinal wheel force; model validation



## 1. Introduction

Fuel consumption is a performance indicator of vehicles. During the design stages, it is necessary to perform analyses capable of determining the output of the future vehicle. Simulation models range from simple to complex, depending on the available information about the components (systems and sub-systems) used for the specific vehicle. When it comes to performing fuel consumption exhaust gases analysis, the common practice is to evaluate the performance using a drive cycle [1]. The specialist worked to construct a Worldwide harmonized Light duty Test Cycle (WLTC) and the corresponding Test Procedure.

The development of the WLTC test cycle was based on real-life traffic data collected from various regions of the world. The data were analyzed, and a series of weighting factors were applied to provide a representative solution for each region investigated [2].

The European Commission replaced the NEDC drive cycle with the WLTP cycle to improve vehicle pollutant emissions or the vehicles' fuel economy [3]. Consequently, the new cycle aims to close the gap between real-life and type-approval $CO_2$ emissions. This is an important issue when fuel consumption is evaluated, as the acceleration phases may have significant importance. Past research revealed that the factors of acceleration vs. speed may require some attention [4,5]. The vehicle emissions were investigated in the case of gasoline, biogas, and biogas plus hydrogen fuels, and both NEDC and WLTP show similar performances [6]. The differences between the implementation of the current drive cycle (WLTP) and the previous version (NEDC) were investigated, including the evaluation of the road loads [7]. It should be noted that research activities involve evaluating the fuel consumption for conventional and plug-in hybrid vehicles that use the NEDC cycle [8].

Procedures have been developed to investigate the entire vehicle or its sub-systems. Starting from the engine, a general-purpose model was proposed [9]. It should be noted that the efforts are aimed towards simplifying the simulation models. The transmission, especially the speed and gear ratio, are an important set of parameters capable of improving fuel consumption performance, especially for hybrid vehicles [10]. The overall output of the

power sources should correlate with the required tractive force. The road load was also used as an input parameter for the development of the control strategy for a P2 Hybrid Electric Vehicle [11]. The multiple power sources or multiple driving axles can only be correlated when the capable driving torque agrees with the existing road loads [12]. Rapid assessment of performance is a tool that specialists focus on during preliminary design stages [13]. The hybrid vehicle [14] provides a solution for fuel economy, considering the regeneration sequences available during deceleration phases and downhill driving [15]. The simulation techniques [16] and tools are in continuous expansion, considering the advancements in numerical analysis. The existing simulation technologies can be further enhanced using deep learning strategies for improved accuracy of the results [17]. Convolutional Neural Networks can be used to model the transient regime for a more realistic analysis of the quality of exhaust gases.

The existing simulation procedures available for light vehicles are tested to be extended [14] for commercial vehicles to evaluate performances. A solution for a hybrid commercial vehicle was investigated using the use of an electric engine over an internal combustion engine as a first approach, while the second approach followed a conventional hybrid functioning scheme [18]. The delivered torque was investigated and compared to the required driving conditions. It was found that the existing hybrid solution is valid and provides satisfactory results. The challenge of developing an accurate driving cycle is still ongoing, and new solutions have been proposed using data-on-board [19]. The conventional driving cycles are also applied to fully electric vehicles [20] as a validation procedure for component performance. The acceleration regime and the modeling solution are essential for estimating vehicle performance [21]. The energy savings are closely related to the braking regimes, and based on these conditions, energy management algorithms can be improved [22,23]. The required torque gives the engine efficiency and affects the performance of the battery charging process [24]. The potential of other regenerative power sources (e.g., solar energy) [25] is evaluated, and the impact of implementing these solutions on vehicle performances can be investigated using simulation models. The solar energy and the work provided by this energy source can be injected into the main power stream [26]. Route planning for maximum efficiency of solar energy is a solution for future transportation technology [27].

Based on the current finding in the field, there are two technical approaches to solving the numerical problem [28]. The first solution is defined as "backward" (or "backward-facing") in which the input is provided by the driving cycle [29]. The required force to drive the cycle is determined based on the vehicle definition. The second solution, using a forward-facing method, is more complex and requires a logical element representing the driver. The "driver" controls the commands of the vehicle accelerator, brake, and gear selector. This logical component should always evaluate the difference between the output, representing the kinematic and dynamic state of the vehicle, and the input, represented by the prescribed speed of the vehicle. Thus, besides the complexity of modeling the mechanical system, developing a correct model for the "driver" is necessary.

Although apparently simplistic, the backward-facing modeling solution is currently implemented to provide an evaluation of $CO_2$ emissions and fuel consumption for light-duty [30] and heavy-duty vehicles [31]. Advanced simulation tools can be designed in open-architecture solutions, allowing the user to add various components according to the desired configuration of the vehicle.

The investigated literature reveals that an accurate estimation of the road loads is required. This way, either torque or power can be determined, and thus, fuel consumption can be estimated. No matter the complexity of the model or the simulation strategy, fuel consumption is related to the tractive force. Therefore, it is possible to have a simple model capable of producing reliable results without considering a complex vehicle construction analysis. The model can be applied to alternative power sources (e.g., solar energy), and can be used to balance the cargo load and the vehicle mass for optimal commercial output.

This paper's backward-facing modeling determines the longitudinal wheel force required to drive the vehicle along a prescribed cycle. Some parameters (e.g., coefficient of rolling resistance, factor accounting for rotational masses) required to estimate the road loads are investigated. Fuel consumption is the key parameter investigated in this paper in terms of the necessary to complete the driving cycle and potential savings considering the deceleration sequences.

Complex MATLAB/Simulink models were used to validate the results. The first model models a conventional vehicle, while the second aims to evaluate a hybrid (mild) vehicle.

A series of parameters were extracted from these models (e.g., drag force, inertia force, rolling resistance) and compared with the results of the analytical model.

This can be seen as an open-architecture solution. The wheel torque can be easily estimated and, by adding transmission efficiency maps for sets of gear ratios, can be transferred to the power source(s).

The longitudinal wheel force is, thus, a given parameter that should be achieved no matter how the drivetrain and logic of the vehicle are constructed.

## 2. Dynamics of the Vehicle in Longitudinal Motion

The vehicle's modeling process begins with defining the basic dynamics. The drive cycle defines a driving scenario that considers the vehicle in longitudinal motion. Thus, the dynamics of the vehicle are substantially simplified. The following equation expresses the equation of motion in the longitudinal direction:

$$F_{W,X} = R_r + R_a + R_g + \frac{W}{g} \cdot \gamma \cdot a \tag{1}$$

The terms listed in Equation (1) [32] represent the total longitudinal wheel force ($F_{W,X}$), total rolling resistance of the vehicle ($R_r$), the aerodynamic resistance ($R_a$), grade resistance ($R_g$), and the acceleration term ($a$), gravitational acceleration ($g$), vehicle weight ($W$), and a coefficient ($\gamma$) which accounts for the rotational masses' influence on the inertia force's translational effect.

The term total longitudinal force is preferred in the paper over the term tractive force. It is considered that the tractive force [32] is the resultant force at the wheel hub. In this case, a minimum force equal to the rolling resistance is required to produce longitudinal displacement of the vehicle.

The rolling resistance force accounts for the effects of the interaction between the pneumatic tire and the surface of the road. It is generally expressed in terms of the vehicle's weight ($W$), and a non-dimensional parameter called the coefficient of rolling resistance ($f_r$).

$$R_r = f_r \cdot W \cdot \cos \alpha \tag{2}$$

The coefficient of rolling resistance can be expressed as a constant value or as a function of the speed of the vehicle [32]:

$$f_r = 0.01 \cdot \left(1 + \frac{V}{160}\right) \tag{3}$$

or:

$$f_r = \left(\mu_s + \mu_k \cdot \frac{V}{3.6}\right) \tag{4}$$

where $\mu_s$ and $\mu_k$ represent the static and kinetic coefficients of rolling resistance ($\mu_s = 1.2 \cdot 10^{-2}$ and $\mu_k = 8.8 \cdot 10^{-5}$ [33]), respectively. For all the functions mentioned above, the speed of the vehicle ($V$) is expressed in km/h.

The aerodynamic resistance $(R_a)$ is usually expressed in the form:

$$R_a = \frac{1}{2} \cdot \rho_{air} \cdot C_D \cdot A_f \cdot v^2 \tag{5}$$

where $\rho$ is the air density $(\rho_{air} = 1.225 \text{ kg/m}^3)$, $C_D$ is the coefficient for the aerodynamic resistance, $A_f$ is the frontal projected area of the vehicle, and $v$ is the speed of the vehicle relative to wind.

The grade resistance represents the longitudinal component of the vehicle and, in this case, is expressed as:

$$R_g = W \cdot \sin \alpha \tag{6}$$

Parameter $\alpha$ represents the longitudinal inclination of the road and is normally expressed as $p_\%$ representing the road grade. For a value of grade up to 15% it can be considered that $\tan \alpha \cong p_\%$ ($\sin \alpha \cong p\%$; $\cos \alpha \cong 1$).

The coefficient $(\gamma)$ used for the inertia force can be expressed as a function of the transmission's ratios $(\xi_0)$ of the final drive and $\left(\xi_{gb}\right)$ of the gearbox [32]:

$$\gamma = 1.04 + 0.0025 \cdot \left(\xi_0 \cdot \xi_{gb}\right)^2 \tag{7}$$

This solution requires the definition of the transmission of the vehicle. Peng et al. [33] provide a variant method to determine this parameter as a function of the vehicle speed:

$$\gamma = \frac{1}{1.02 - \frac{5.22}{V}} \tag{8}$$

Equation (1) can be written as:

$$F_{W,X} = f_r \cdot W + \frac{1}{2} \cdot \rho \cdot C_D \cdot A_f \cdot v^2 + p_\% \cdot W + \frac{W}{g} \cdot \gamma \cdot \frac{dv}{dt} \tag{9}$$

The following layout is preferred to reflect the performance of the vehicle:

$$G - p_\% = \frac{\gamma}{g} \cdot \frac{dv}{dt} \tag{10}$$

and includes the measure of the gradeability $(G)$:

$$G = \frac{F_T - R_a - R_r}{W} \tag{11}$$

## 3. Backward-Facing Simulation of Vehicle Dynamics

In the backward-facing simulation, the dataset defining the specific drive cycle is used to compute, by a set of empirical equations, the necessary wheel longitudinal force $(F_{W,X})$. The analysis model is simplified, as there is no need for a driver model or a reactive loop to control the drivetrain parameters (Figure 1).

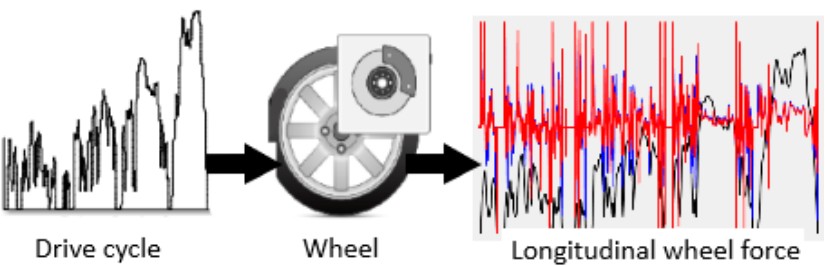

**Figure 1.** Backward-facing modeling strategy.

The cycle data are generally defined as a collection of points defining the vehicle's time and required speed (in some cases, the gradient can be defined). Figure 2 presents the speed and acceleration versus time for a specific drive cycle (e.g., Worldwide Harmonized Light-Duty Vehicles Test Procedure (WLTP)).

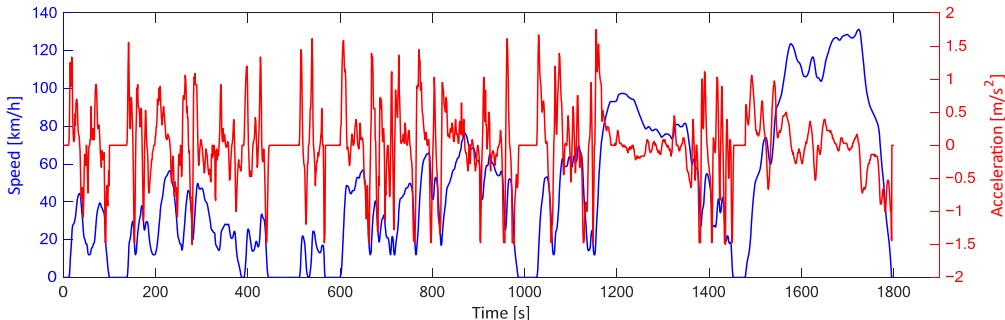

**Figure 2.** Drive cycle (WLTP, Class 3). Speed and acceleration vs. time.

The space traveled during the drive cycle is presented in Figure 3.

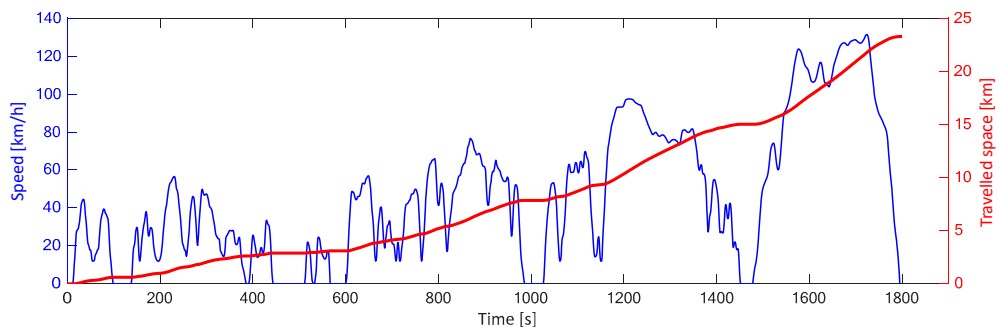

**Figure 3.** Drive cycle (WLTP, Class 3). Speed and traveled space vs. time.

The work of the rolling resistance $(W_{r_i})$ is estimated using the following Equation:

$$W_{r_i} = R_{r,i} \cdot \Delta d_{i,i-1} \cdot \text{sgn}(v_i) \tag{12}$$

The sign operator (sgn) was introduced to account for the intermediate phases $(V_i = 0 \text{ km/h})$ that are identified in the drive cycle. Otherwise, a positive value of the rolling resistance is computed, even if the vehicle is stationary. The distance travelled during the reference time interval from time $t_{i-1}$ to time $t_i$ is denoted by $\Delta d_{i,i-1}$.

The work of the aerodynamic drag $(W_{a_i})$ is estimated by the formula:

$$W_{a_i} = R_{a,i} \cdot \Delta d_{i,i-1} \tag{13}$$

while the work consumed to accelerate (restored during deceleration phases) the vehicle $(W_{i_i})$ is estimated by:

$$W_{i_i} = \frac{W}{g} \cdot \gamma \cdot \frac{dv}{dt} \cdot \Delta d_{i,i-1} \tag{14}$$

Figure 4 presents the results obtained for the current driving cycle. The results are cumulated for each reference time interval to report the instantaneous energy required to drive the vehicle. The reference time interval represents the time between two entries in the driving cycle dataset.

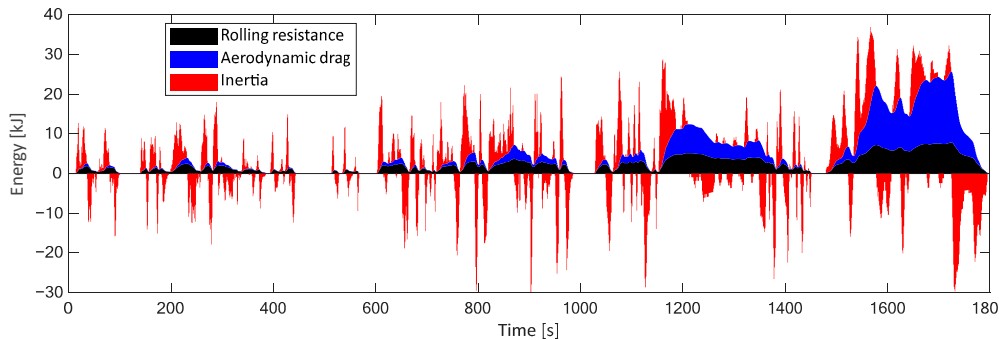

**Figure 4.** The work of the road loads (reference time interval values).

The results can be conveniently manipulated to provide a representation of the energy that can be recovered $(W_{i_i})$ [34] during the deceleration phases of the vehicle (Figure 5). It should be noted that the effect of the permanent road loads $(W_{r_i} + W_{a_i})$ was subtracted from the available (recovered) work.

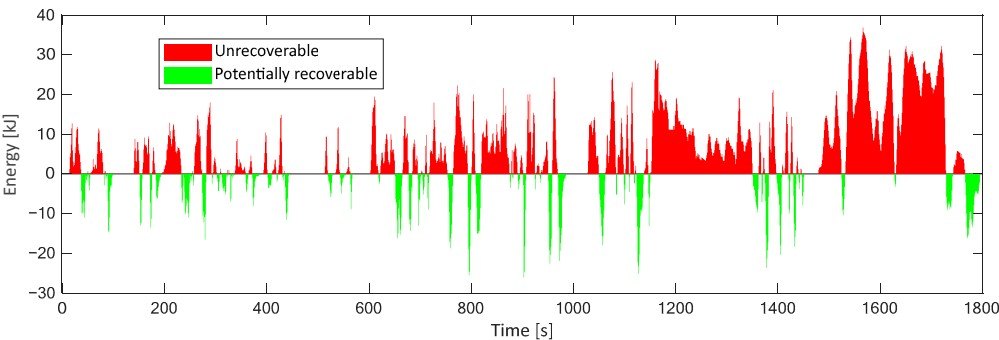

**Figure 5.** Unrecoverable vs. potentially recoverable work (reference time interval values).

The values can be cumulated for each reference time interval to display the total energy required to drive the vehicle. The energy that might be potentially recovered during the deceleration phases is thus determined (Figure 6).

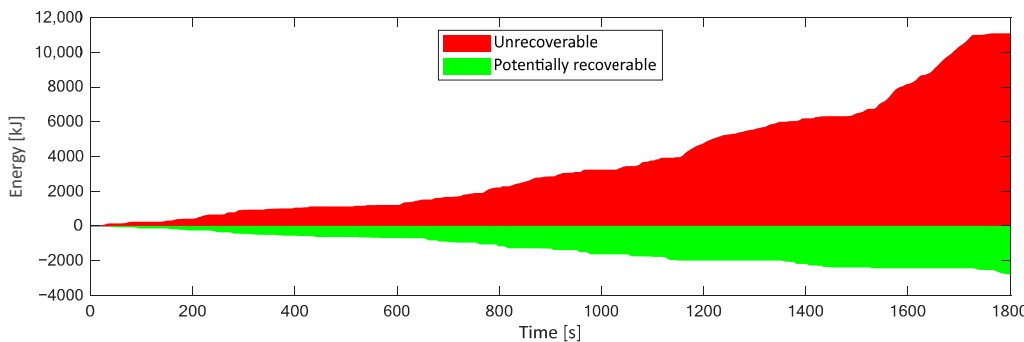

**Figure 6.** Total unrecoverable vs. potentially recoverable work.

## 4. Forward-Facing Simulation of Vehicle Dynamics

Forward-facing simulation is more complex and is designed to encapsulate the logic necessary to control the powerplant and driveline of the vehicle, providing the longitudinal wheel force necessary to overcome the resistance forces as defined by the drive cycle. For the current analysis, a MATLAB/Simulink model was used. This model is pre-defined as a "Conventional Vehicle Reference Application". Figure 7 presents the general configuration of the simulation model.

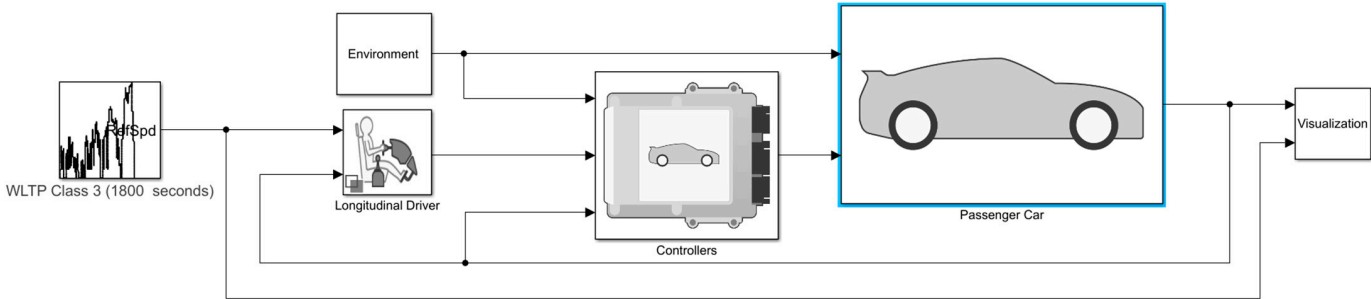

**Figure 7.** The general configuration of the MATLAB/Simulink mode (Conventional Vehicle Reference Application).

The simulation model is complex, and allows a number of variants to be defined. The SimpleEngine sub-system was selected using the Variant Manager to define a compatible model with the one defined by the set of analytical equations. A view of the sub-systems is presented in Figure 8.

| Name | Submodel Configuration | Variant Control | Condition |
|---|---|---|---|
| SiCiPtReferenceApplication | | | |
|   Controllers | | | |
|   Drive Cycle Source | | | |
|   Longitudinal Driver | | | |
|   Passenger Car | | | |
|     Drivetrain (Submodel: DrivetrainConVeh) | | | |
|       DCT | | | |
|       Differential and Compliance | | | |
|         All Wheel Drive | | 2 | (N/A) |
|         Front Wheel Drive | | 0 | (N/A) |
|         Rear Wheel Drive | | 1 | (N/A) |
|       Vehicle | | | |
|         Vehicle Body 3 DOF Longitudinal | | | |
|       Wheels and Brakes | | | |
|     Engine | | | |
|       CiEngine (Submodel: CiEngine) | | CiEngine | (N/A) |
|       CiMappedEngine (Submodel: CiMapped... | | CiMappedEngine | (N/A) |
|       SiDLEngine (Submodel: SiDLEngine) | | SiDLEngine | (N/A) |
|       SiEngine (Submodel: SiEngine) | | SiEngine | (N/A) |
|       SiMappedEngine (Submodel: SiMapped... | | SiMappedEngine | (N/A) |
|       SimpleEngine (Submodel: SimpleEngine) | | SimpleEngine | (N/A) |

**Figure 8.** Variant Manager application showing the configuration of the Conventional Vehicle.

The Conventional Vehicle can be extended to allow the use of an electric propulsion system using the motor, generator units, and an energy storage unit (batteries pack).

Figure 9 presents a generic configuration of the vehicle used for the forward-facing simulation. The symbol ⇨ colored in red defines the power flow for the Conventional Vehicle, while the green arrow adds the electrical components to the driveline.

The parameters used for the simulation models (forward-facing simulation) and the theoretical analysis (backward-facing model) are listed in Table 1.

The current analytical model uses the engine and transmission's global efficiency parameters (constants). The efficiency parameters were extracted from the MATLAB/Simulink models after performing the power and energy analysis. The engine and transmission efficiency values can be determined from the experimental data, complex simulation model, or evaluated as averaged data.

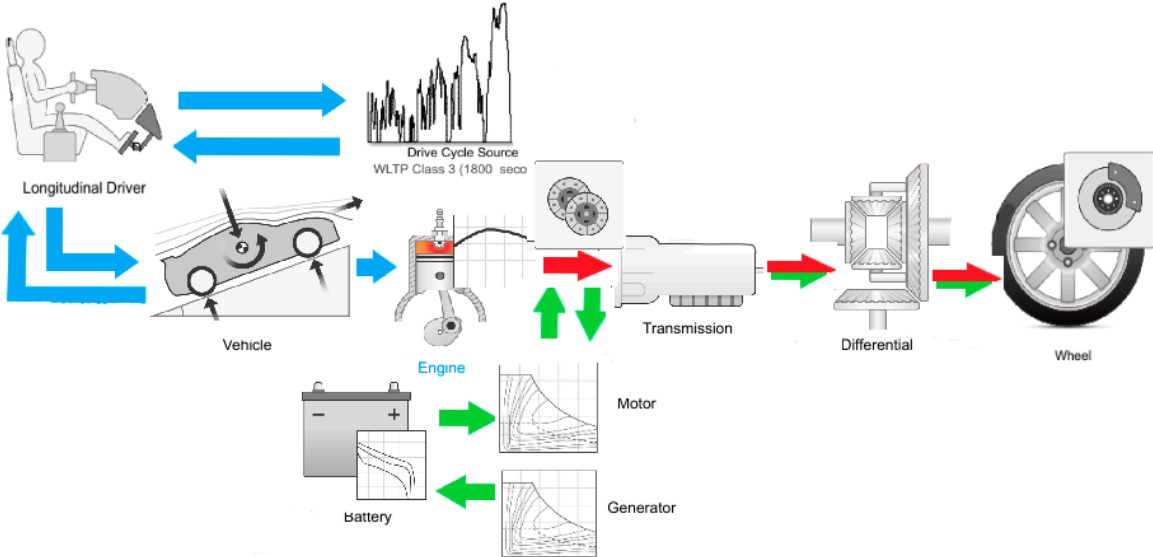

**Figure 9.** Forward-facing simulation model. The virtual driver follows the drive cycle by controlling the acceleration, brake, and gear selector input commands. The vehicle's response is monitored and constantly adapted to the requirements of the drive cycle. The vehicle's conventional (autoblkCon-VehStart) and hybrid (autoblkHevStart) configurations are presented.

**Table 1.** Parameters for the theoretical and simulation models.

| Parameter(s) | Symbol(s) and Unit(s) | Value |
|---|---|---|
| Theoretical Models | | |
| Vehicle mass | $m$ [kg] | 1200 |
| Drag coefficient | $C_D$ [−] | 0.3 |
| Frontal area | $A_f$ [m$^2$] | 2.0 |
| Engine efficiency | $\eta_{ICE}$ [−] | 0.24 |
| Driveline efficiency | $\eta_{DL}$ [−] | 0.92 |
| Energy conversion efficiency | $\eta_{ME}$ [−] | 0.65 |
| Fuel density (E10) | $\rho_{E10}$ [kg/m$^3$] | 754 |
| Fuel (gross) calorific value (GCV) | $GCV_{E10}$ [MJ/kg] | 46.4 |
| MATLAB Models | | |
| Brake-specific fuel consumption | $BSFC$ [g/kWh] | 275 |
| Battery rated capacity | [Ah] | 5.2 |
| Battery State of Charge | [−] | 80% |
| Generator–max. torque | [Nm] | 100 |

The brake-specific fuel consumption is a performance parameter of the internal combustion engine ($BSFC$), and is highly dependent on the functioning regime of the engine.

For the MATLAB/Simulink model, the $BSFC$ was constrained to a constant value to match the hypothesis of constant efficiency parameters used for the analytical solution. This constant can be seen as a target value to be achieved using detailed analyses of the driveline parameters (final drive ratio, gear ratios, efficiency).

The first iteration was performed using the conventional vehicle model. Figure 10 presents the longitudinal wheel force computed using both the analytical model and as a result of the MATLAB/Simulink analysis.

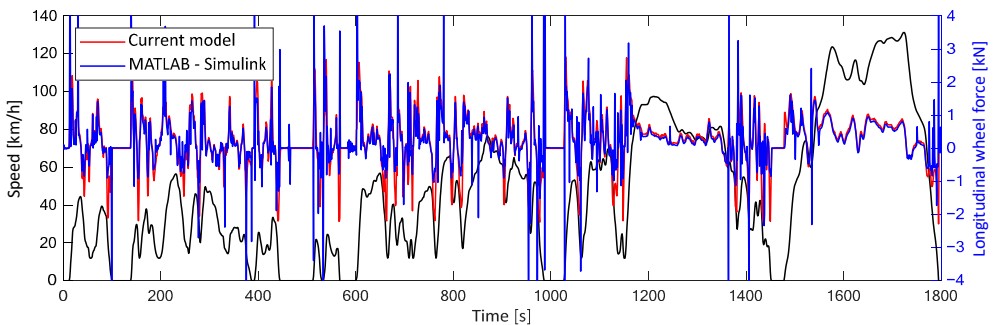

**Figure 10.** Longitudinal wheel force ($F_{W,X}$): backward- vs. forward-facing analysis.

Results presented in Figure 8 are in good agreement, showing that the analytical solution can be successfully used to identify the wheel force necessary to overcome the road loads.

## 5. Fuel Consumption Analysis

The most important parameter related to vehicle performance is fuel consumption. This parameter can be directly related to the quantity of pollutant emissions, although these issues should be particularly addressed considering the technical solutions used for the treatment of exhaust gases.

A convenient approach to evaluate a vehicle's fuel consumption is to convert the work required to drive the vehicle into heat.

The proposed solution is very flexible and can be applied to various fuels once the density and calorific value are determined. It can also be used to perform economic analyses, considering the price of the fuel for increased efficiency.

Thus, the following equation is obtained to determine the fuel consumed during the reference time interval:

$$Q_i = \frac{W_i \cdot 1000}{\rho_{fuel} \cdot GCV \cdot \eta_{ICE} \cdot \eta_{DL}} \tag{15}$$

The total work ($W_i$) is defined as:

$$W_i = W_{r_i} + W_{a_i} + W_{i_i} \tag{16}$$

The reference fuel consumption $Q_{l/100km}$ is evaluated from the drive cycle as:

$$Q_{l/100km} = Q_i \cdot \frac{100\ [km]}{d_{cycle}\ [km]} \tag{17}$$

Figure 11 presents the fuel consumption results obtained using the analytical solution and the simulation model.

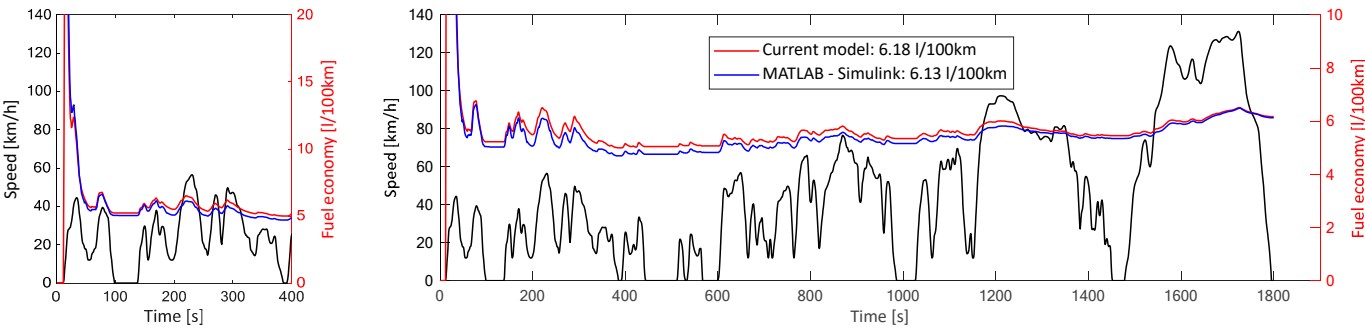

**Figure 11.** Fuel consumption analysis for the conventional vehicle model ((**left**): detailed view of the low-speed sequence; (**right**): complete driving cycle).

The analytical model estimates a fuel consumption ($Q_a$) of 6.18 L/100 km, while the simulation model ($Q_s$) predicts a value of 6.13 L/100 km. The difference can be expressed using the formula:

$$diff\,\% = \frac{Q_s - Q_a}{Q_s}\cdot 100\% \tag{18}$$

Using the above-mentioned results, the difference between the results is $-0.69$ %, thus showing a slight increase in the value reported using the analytical model. Additional results obtained for different driving cycles are presented in Appendix A.

Further improvement of the reported fuel consumption can be achieved by considering the simulation models using steady-state engine maps [11].

Figure 4 reports some work that can be potentially harvested and converted into wheel force if electrical equipment is fitted on the vehicle. The electric equipment must have the capability to provide generative braking, store energy, and drive the vehicle. Although this additional equipment must be fitted on the vehicle, it was assumed that the vehicle mass of the hybrid vehicle is identical to the mass of the conventional vehicle. The battery capacity was limited to 5.2 Ah to agree with the above-mentioned hypothesis.

The work ($W_{i,s}$) that can be potentially recovered from the wheel is defined as:

$$W_{i,s} = W_{i_i} - (W_{r_i} + W_{a_i}) \tag{19}$$

Figure 12 presents the fuel consumption results obtained using the analytical solution and the simulation model, in the case of the hybrid vehicle.

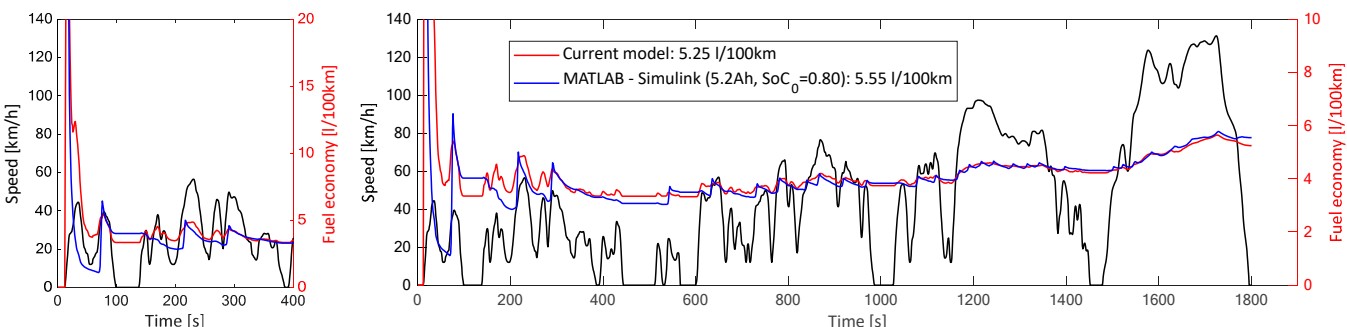

**Figure 12.** Fuel consumption analysis for the conventional vehicle model ((**left**): detailed view of the low-speed sequence; (**right**): complete driving cycle). The electrical component of the propulsion system results in a decrease in fuel consumption, which can be observed at the beginning of the driving cycle when the battery is charged at the nominal capacity (80%).

The analytical model estimates a fuel consumption of 5.25 L/100 km, while the simulation model predicts a value of 5.55 L/100 km. The difference is 5.45 %, in favor of the analytical solution, probably due to the logic of the controller of the electrical system. Further enhancements in the logic of the control module can improve the results [35,36].

In this case, for the analytical solution, the quantity of saved fuel ($Q_{i,s}$) was estimated using Equation (19) and considering that the energy is generated during braking, stored, and converted into mechanical energy. Thus, the fuel saved can be estimated using the following Equation:

$$Q_{i,s} = \frac{W_i\cdot 1000}{\rho_{fuel}\cdot GCV\cdot \eta_{ICE}\cdot \eta_{DL}}\cdot \eta_{ME}\cdot \eta_{DL} \tag{20}$$

The results of the current model can be further calibrated using advanced analytical models developed primarily to work with publicly available data [37]. The models are defined as Power-Based Fuel Consumption and can be integrated into the current analytical solution.

## 6. Discussion

There are advantages to each of the two methods presented in this paper. The backward-facing simulation can be conveniently implemented in various programming languages and spreadsheet applications. However, each additional parameter that is implemented in the application should be defined and related to the currently implemented solution.

The forward-facing model is complex and capable of providing multiple points of information related to the performance of systems integrated into the vehicle model, but it requires some effort to develop and validate before using the outputs. Figure 13 presents a drivetrain sub-system of the vehicle model, showing a series of additional outputs.

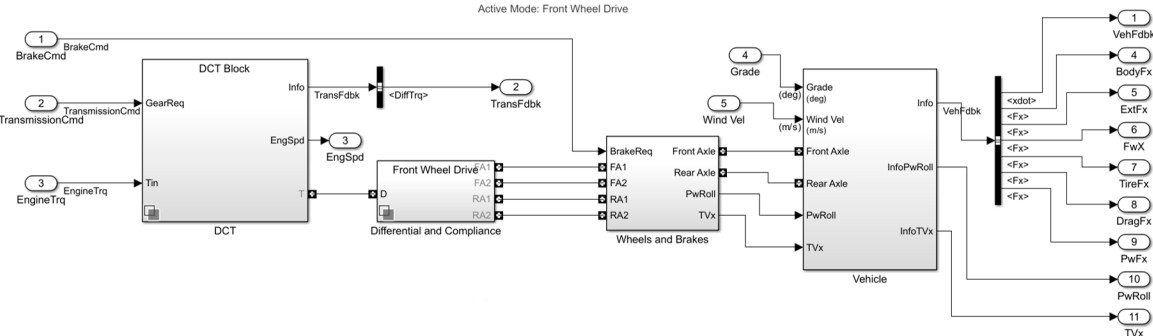

**Figure 13.** The drivetrain sub-system. Additional outputs were selected from the Info Bus of the vehicle model (body, aerodynamic, and tire forces).

The supplemental outputs were subsequently defined in the data flow path to be finally displayed and saved for further analyses. Figure 14 presents the definition of the outputs of the additional data.

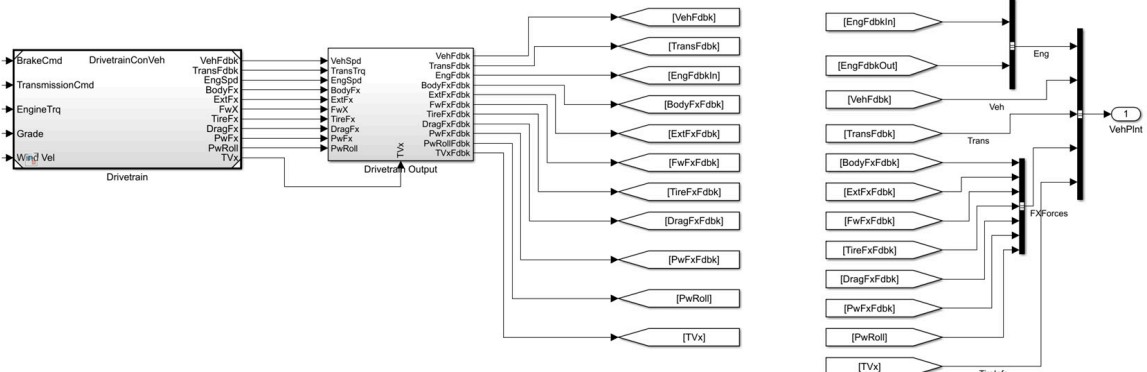

**Figure 14.** Definition of the outputs for the additional model data.

This work demonstrates that a combination of the two can be a successful solution for the development of a vehicle simulation model.

When fuel consumption is the focus of the analysis, the backward-facing model, as demonstrated, is capable of providing accurate results. Furthermore, as demonstrated by other technical solutions, e.g., embedding details maps of the parameters, even exhaust gases can be analyzed to determine the quality and quantity of pollutants.

To complete the present work, a comparison of the components involved in the evaluation of the total longitudinal force is included.

In the case of the backward-facing model, the rolling resistance was evaluated using Equation (2), with the rolling resistance coefficient defined by Equation (3). The Simulink model uses a more advanced solution involving an Equation, namely "Magic Formula",

determined by Pacejka [38]. The equation involves the vertical wheel load ($F_z$) that is not a constant value, and it is constantly affected by the dynamic regime of the vehicle.

Figure 15 presents a comparison of the results obtained for the rolling resistance. The results reported in the figure were extracted from available MATLAB/Simulink datasets:

$$R_{r,s} = \frac{\text{TransTrq}}{r_r} - \text{FwX} \tag{21}$$

where TransTrq parameter is determined from the drivetrain block, FwX is the longitudinal wheel force, and $r_r$ is the wheel rolling radius ($r_r = 0.30\,[\text{m}]$).

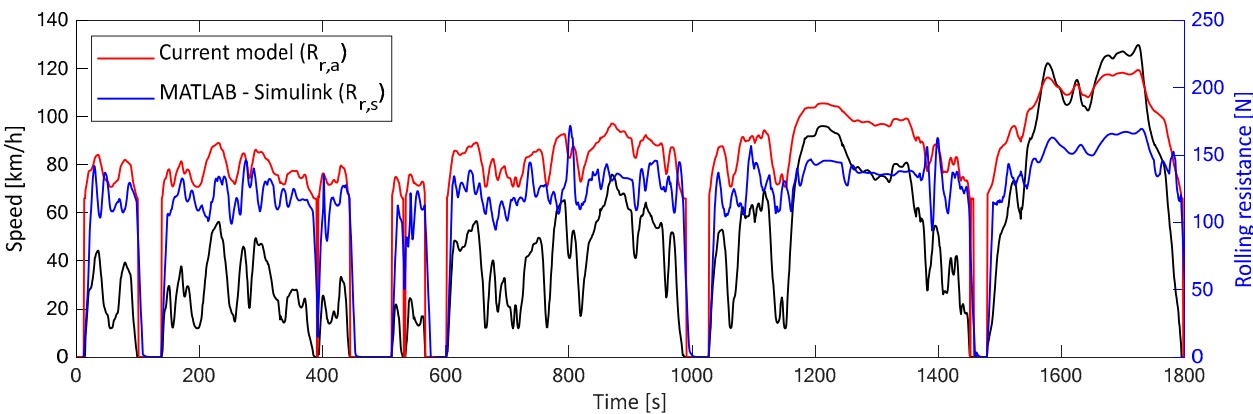

**Figure 15.** Results for the rolling resistance (rolling resistance coefficient is defined by Equation (3)).

It can be noticed that there are some differences between the values, although considering the overall longitudinal wheel force, these differences do not significantly alter the results. The difference is about 50 N, about 5% of the longitudinal wheel force (see Figure 10, time 1600 s).

A different selection for the estimate of the rolling resistance coefficient, namely Equation (4), slightly underestimates the rolling resistance (Figure 16).

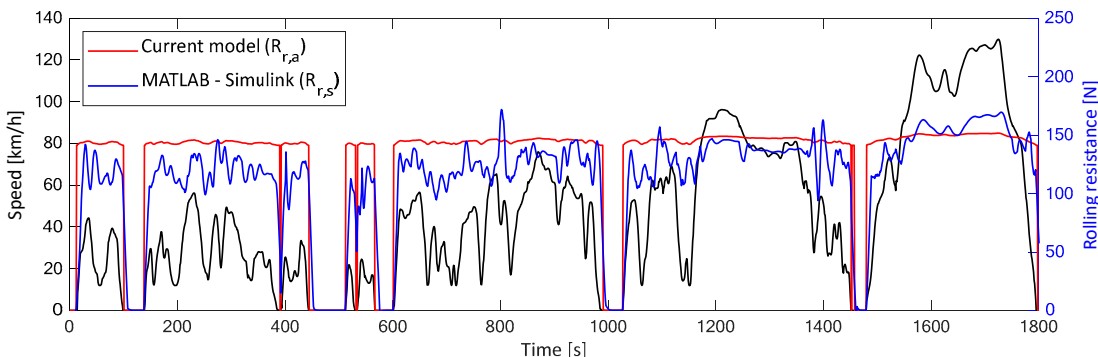

**Figure 16.** Results for the rolling resistance (rolling resistance coefficient is defined by Equation (4)).

The aerodynamic drag is rather straightforward to evaluate [39]. Equation (5) reveals that it can be computed using a series of parameters and the velocity of the vehicle.

Figure 17 presents a comparison of the results obtained for the aerodynamic drag.

In the case of the backward simulation, the inertia is estimated using the translational mass of the vehicle and the acceleration. The translational mass of the vehicle is a specific term, including the effect of rotating masses. The conversion can be performed using the technique of equivalent mass.

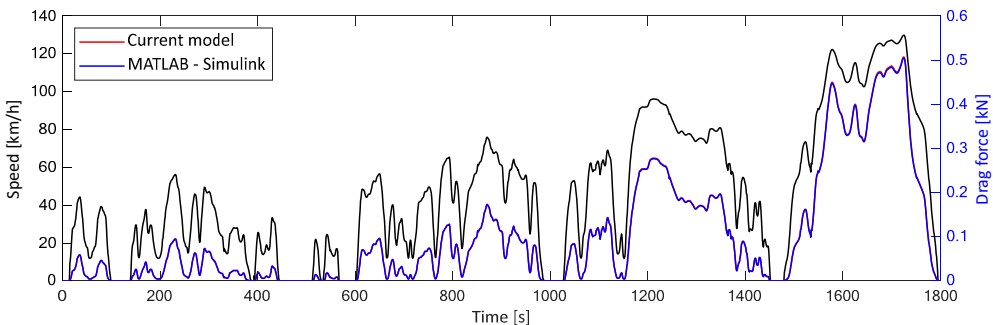

**Figure 17.** Results for the aerodynamic drag (results are identical as the same mathematical model is used for the evaluation of the aerodynamic drag).

This paper defines the longitudinal mass by multiplying the vehicle mass with a factor defined by Equation (8). To avoid high dynamic effects during acceleration and braking in the vicinity of zero value, the speed parameter was defined as:

$$V = \max(V_i,\ 20) \qquad (22)$$

The reference speed (e.g., 20 [km/h]) can be adapted considering various transmission configurations and shifting performances.

The results were filtered ($\mathrm{dataout} = \mathrm{filter}(p1, p2, \mathrm{datain}); p1 = (1/10)\cdot\mathrm{ones}(1, 10); p2 = 2$) to clearly outline the variation of the inertia during the driving cycle. The MATLAB/Simulink output is BodyFX. Figure 18 presents the results obtained from the analyses.

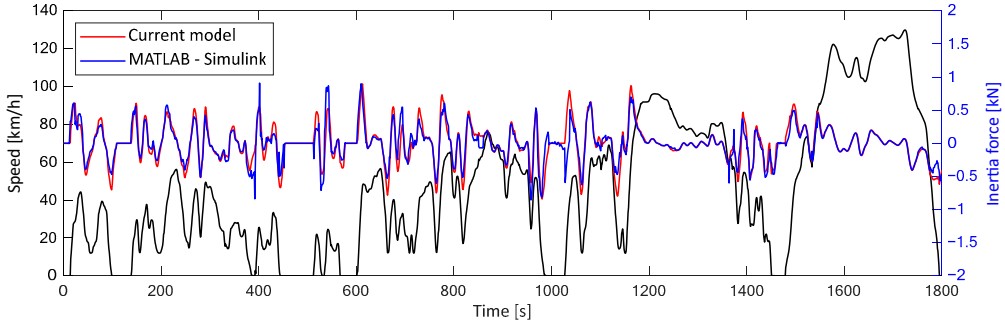

**Figure 18.** Results for the inertia force.

There are some differences between the exact values; however, the paths of the curves are identical. This provides good input for the analysis of the acceleration strategy for the vehicle to minimize fuel consumption [40].

The parameter defined by Equation (8) can be adapted to better capture the influence of the rotating masses. Table 2 presents the values of the parameter for a number of reference speeds (see Equation (22)).

**Table 2.** Parametric analysis of $\gamma$.

| Reference Speed $V$ [km/h] | $\gamma$ Equation (8) | $\left(\xi_0 \cdot \xi_{gb}\right)$ Equation (7) |
|:---:|:---:|:---:|
| 5.5 | 14.102 | 72.28 |
| 10 | 2.008 | 19.677 |
| 20 | 1.317 | 10.536 |
| 30 | 1.182 | 7.537 |

The result of the parameter was subsequently used to determine the product between the final drive ratio and (maximum) gear ratio, using Equation (23).

$$\left(\xi_0 \cdot \xi_{gb}\right) = \sqrt{\frac{\gamma - 1.04}{0.0025}} \tag{23}$$

This can be useful to define a reference speed of Equation (8) considering the initial value of the product $\left(\xi_0 \cdot \xi_{gb}\right)$ between 10 and 14, corresponding to a conventional vehicle configuration. Figure 19 presents the results obtained for different values of the reference speed.

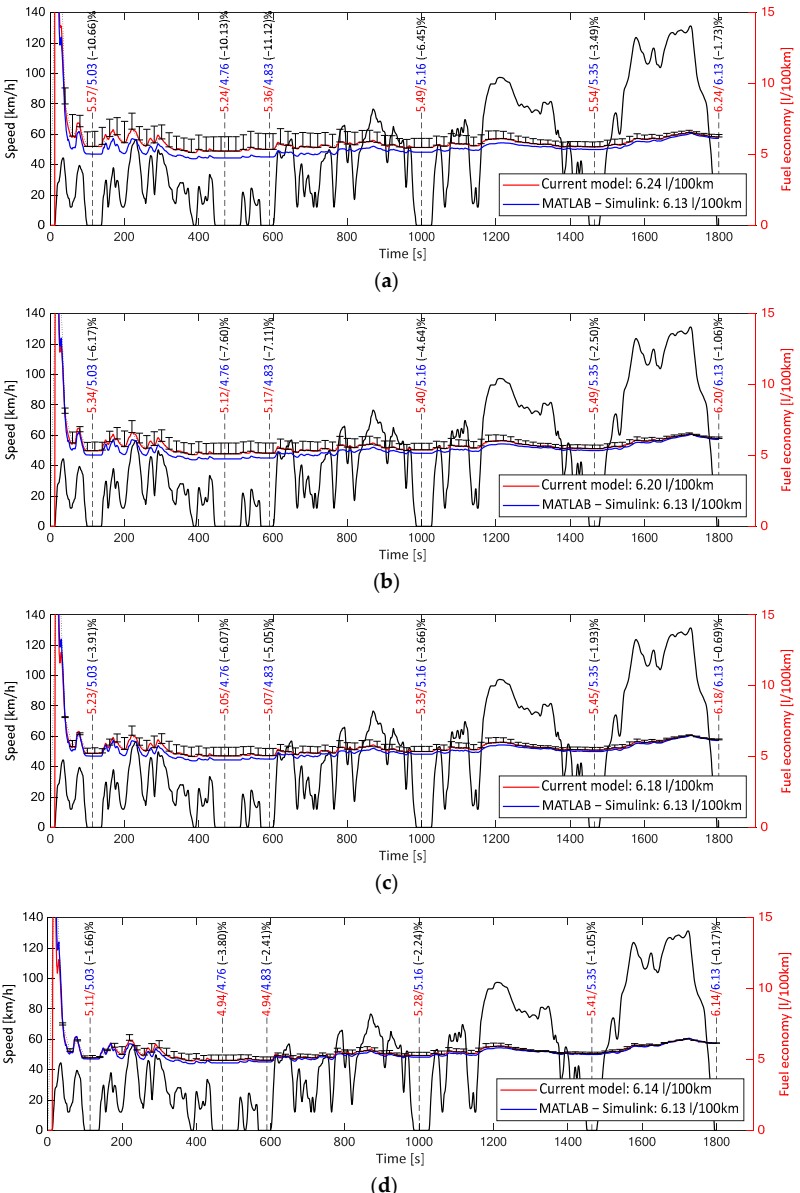

**Figure 19.** The influence of the parameter $\gamma$ on the fuel consumption performance. The estimated and computed fuel consumption numbers were extracted from the analytical and simulation dataset when the engine was idling. Results show that when increasing the reference speed in Equations (8) and (22), the differences are within 5%. Numbers in red report the analytical results, numbers in blue report simulation results, and the differences are computed using Equation (18). A negative number gives a better smaller value for the fuel consumption determined from the MATLAB/Simulink model. (**a**) $V = 5.5$ [km/h]; (**b**) $V = 10$ [km/h]; (**c**) $V = 20$ [km/h]; (**d**) $V = 30$ [km/h].

The error bar is upward when the analytical model results exceed the simulation data.

A simple application of the backward-facing simulation is proposed. The application points to fuel consumption considering a conventional vehicle and a vehicle capable of exploiting the deceleration phases.

The problem is finding the maximum mass of a vehicle that can be constructed as a hybrid with similar fuel consumption to the vehicle analyzed in this paper (mass, drag coefficient, frontal area, transmission configuration, and wheels).

A series of iterations was performed considering the mass of the vehicle ranging from 1200 kg to 1800 kg. Using the results from the backward-facing analysis (Figure 20), the solution was identified as $m = 1650$ kg.

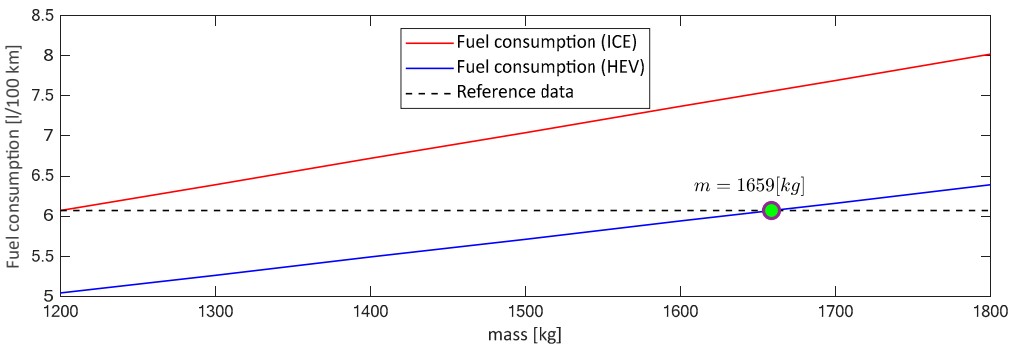

**Figure 20.** Solution for the vehicle mass.

The value was adapted into the forward-facing simulation model of the hybrid vehicle to evaluate fuel consumption. A comparison of the results is presented in Figure 15.

The data presented in Figure 21 indicate the fuel consumption for the reference vehicle with a mass of 1200 kg, the fuel consumption for a hybridized reference vehicle, and the potential mass of a vehicle that will display a fuel consumption similar to the reference vehicle.

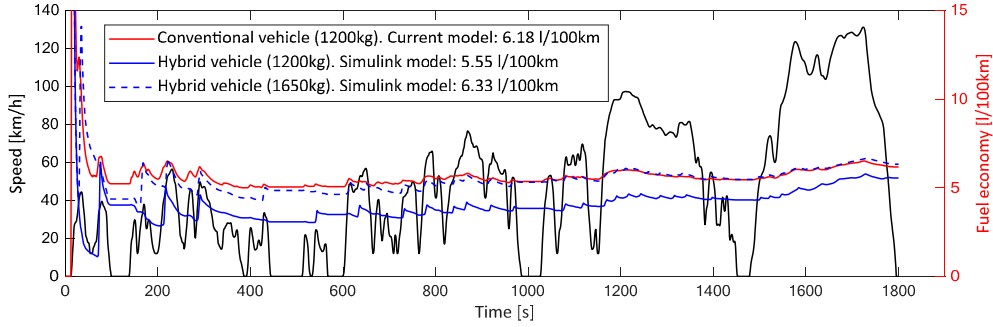

**Figure 21.** Fuel consumption results.

The benefit of this analysis is that it can provide a quick estimate of the influence of the mass increase in fuel consumption, especially when the preliminary design of the vehicle is addressed.

## 7. Conclusions

The paper addresses the longitudinal wheel force and the processes that can be applied to analyze fuel consumption. According to the input dataset, the longitudinal wheel force required to drive along a specific cycle is evaluated at the reference time interval. Furthermore, the work required is evaluated and converted to fuel consumption using some use-defined parameters. At the same time, the longitudinal wheel force analysis can provide information about potential fuel savings. The method is direct and uses the efficiency parameters extracted from the simulation models.

MATLAB/Simulink models were used and investigated to prove the above-mentioned method. It was found that the results are in good agreement, proving the potential of the proposed method as a first step in development.

Furthermore, the individual components of the road loads were investigated. The analysis includes some details related to the selection of the parameters and the influence over the reported fuel consumption. The process of data extraction from the complex simulation models was demonstrated, showing the parameters and manipulation methods. The differences between forward- and backward-facing simulation models were evaluated.

An application related to the vehicle mass is presented. This can be a useful solution when the implementation of different safety systems should be implemented on existing vehicle models.

Results for other driving cycles are presented, reporting intermediate and final values of the fuel consumption. This analysis can be useful for calibrating the parameters used for the analytical model.

These demonstrate the applicability of a simple calculation method capable of supporting additional extensions to simulate internal combustion engines, electric motors, or a combination of the two. It can be easily used to identify some limits (e.g., mass) that are affecting fuel performance.

**Author Contributions:** Conceptualization, S.T. and D.P.; methodology, S.T. and D.P.; software, S.T.; validation, S.T. and D.P.; formal analysis, S.T. and D.P.; writing—original draft preparation, S.T. and D.P.; writing—review and editing, S.T. and D.P. All authors have read and agreed to the published version of the manuscript.

**Funding:** This research received no external funding.

**Informed Consent Statement:** Not applicable.

**Data Availability Statement:** Not applicable.

**Conflicts of Interest:** The authors declare no conflict of interest.

## Appendix A. Supplemental Fuel Consumption Analyses

The results presented in this Appendix A were determined using the MATLAB/Simulink Convetional Vehicle model. For the analytical solution; the inertia force was estimated using a reference speed of 20 [km/h]. The reported values of the fuel consumption show a good agreement between the analytical and simulation datasets (average difference $< 5\%$, while the maximum is reported for NEDC cycle during star phase $\approx 40\%$).

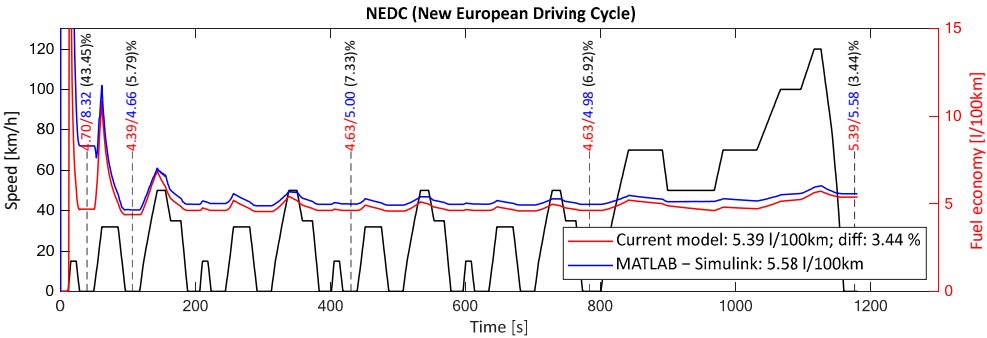

**Figure A1.** Analytical and simulation results for the New European Driving Cycle (NEDC). The NEDC driving cycle (updated 1997) was designed to assess fuel economy in passenger cars. This driving cycle is composed of two sections: the ECE, an urban driving cycle, and EUDC (Extra Urban Driving Cycle) to account for the more aggressive, high-speed driving modes [1].

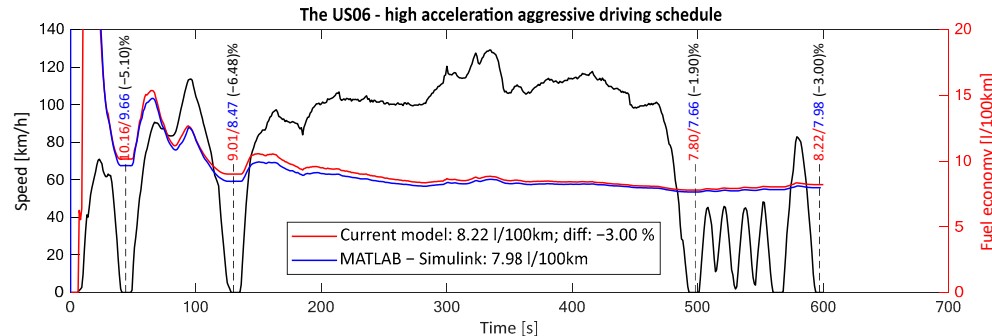

**Figure A2.** Analytical and simulation results for the US06 Supplemental Federal Test Procedure. This driving cycle is designed to represent aggressive, high-speed, high-acceleration driving behavior and rapid speed fluctuations [5].

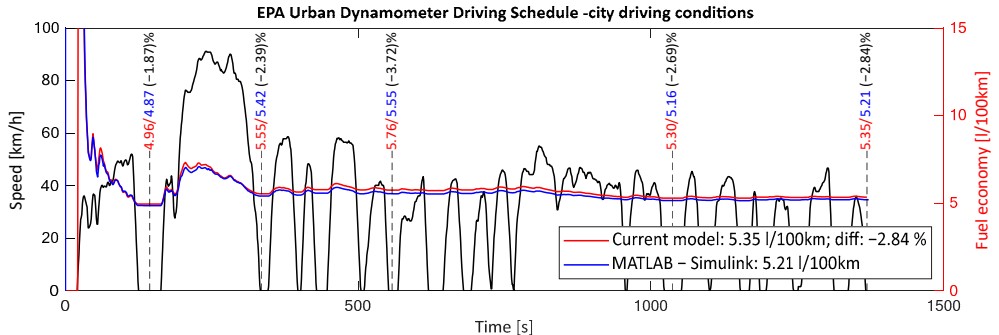

**Figure A3.** Analytical and simulation results for the EPA Urban Dynamometer Driving Schedule. This driving cycle is designed for city driving conditions [1].

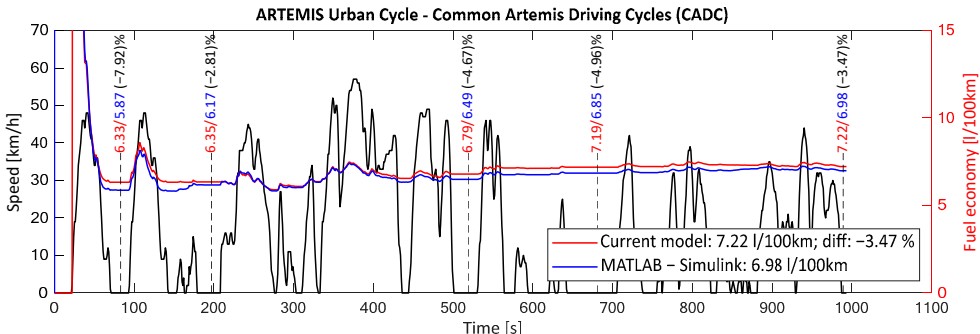

**Figure A4.** Analytical and simulation results for the ARTEMIS urban driving cycle. This driving cycle is part of a project to define a set of harmonized driving cycles using a large database describing actual driving conditions in Europe [41].

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
