# Peer review of "Backward-Facing Analysis for the Preliminary Estimation of the Vehicle Fuel Consumption"

_sustainability, doi:10.3390/su15065344_

Round 1

Reviewer 1 Report

Assessment of fuel consumption for hypothetical traffic scenarios and vehicle operating conditions requires tools in the form of reliable mathematical models estimating fuel consumption based on a set of selected input parameters. The actual fuel consumption occurring during vehicle operation is a function of many factors having varying degrees of impact.  These factors can be aggregated appropriately as factors related to vehicle characteristics and systems; to environmental and traffic conditions and to the driver respectively. In the paper only factors related to vehicle characteristics are considered.

Chapter 1 presents some background but not the state of the art for vehicle fuel consumption models. The discussion of NEDC vs WLTP and the need to develop a representative driving cycle is off topic. Moreover, the novelty of the article is not shown. Chapter 2 describes the already known simulation model, which was used to calculate fuel consumption for driving conditions in accordance with the WLTP driving cycle.

Chapter 3 discusses potential fuel savings.

The reference model was introduced in Chapter 4 but without the details that are necessary to recognize the overall complexity of the model. This chapter should be extended. The results of the fuel consumption analysis are presented in chapter 5. The authors proposed a comparison based on single values ​​of Qa and Qs (Qa and Qa should be described in the text as analytical and simulated values, respectively).

Simple discussion is presented in chapter 6 (wrongly numbered as 5-th). I disagree with statement that results obtained for rolling resistance shows marginal differences. One can see that for speed exceeding 80 km/h  the differences of rolling resistance can be about 50 kN (c.a. 25%). At the end of the chapter simple application of the model is proposed.

In my opinion, the authors should rewrite the manuscript. It will be valuable to show the accuracy of the model for a real traffic scenario (i.e. non-zero road gradients) compared to real measurement data.

1. What is the main question addressed by the research? The work deals with the issue of estimating fuel consumption for given conditions of vehicle motion. 2. Do you consider the topic original or relevant in the field? Does it address a specific gap in the field? The problem is well known, as is the main method used by the authors (see e.g. https://doi.org/10.1016/j.trd.2011.05.008; https://doi.org/10.3390/en14237979. The model is one of the simplest models even among the black box models. Unfortunately, I do not identify the knowledge gap filled by the results of this paper 3. What does it add to the subject area compared with other published material? A comparison of the results obtained using the simple model and the MATLAB model is given. This is the most interesting part of the article. 4. What specific improvements should the authors consider regarding the methodology? What further controls should be considered? The necessary list of improvements is as follows: -comparison of the result obtained from the model with the actual fuel consumption rate in the WLTP cycle for the vehicle under consideration -detailed description of the MATLAB model to recognize the overall complexity of the model and its shortcomings, -more extensive comparison of the analytical model and MATLAB in terms of the calculated force on the wheel and the use of appropriate measures to further discuss the differences -the accuracy of the backward model for a real traffic scenario (i.e. non-zero road gradients) compared to real measurement data -an extensive new part in which the authors present examples of the possible practical application of the model 5. Are the conclusions consistent with the evidence and arguments presented and do they address the main question posed? Conclusions should be rewritten to summarize the pros and cons of the backward model. 6. Are the references appropriate? Should be review (discussion about NEDC vs WLTP cycle is off topic) and completed with references to Vehicle Power Specific models 7. Please include any additional comments on the tables and figures. Some explanations (see line 155) and equations (13, 14) are trivial. The same is about Fig. 1. Fig. 7 cannot be treated as a sufficient description of the MATLAB model. Figures 8, 9 and 10-13 should be redrawn to better show the outputs of both models. Figure 13 has an incorrect caption.

Author Response

Dear Reviewer,

We kindly thank you for your time and efforts in reviewing our manuscript.

We are also grateful for the opportunity of submitting a revised version.

Please receive the updated manuscript following the comments.

Thank you!

Best wishes!

Comment: Assessment of fuel consumption for hypothetical traffic scenarios and vehicle operating conditions requires tools in the form of reliable mathematical models estimating fuel consumption based on a set of selected input parameters. The actual fuel consumption occurring during vehicle operation is a function of many factors having varying degrees of impact. These factors can be aggregated appropriately as factors related to vehicle characteristics and systems; to environmental and traffic conditions and to the driver respectively. In the paper only factors related to vehicle characteristics are considered.

Response: Thank you for your comment!

The analysis presented in this paper is designed to be simple to implement to serve as a starting point for detailed work. However, considering the simple architecture of the model allows further development, including a model of the driver and the use of detailed parameters.

Comment: Chapter 1 presents some background but not state of the art for vehicle fuel consumption models. The discussion of NEDC vs WLTP and the need to develop a representative driving cycle is off topic. Moreover, the novelty of the article is not shown. Chapter 2 describes the already known simulation model, which was used to calculate fuel consumption for driving conditions in accordance with the WLTP driving cycle.

Response: Thank you for your comment!

The section is indeed dedicated to a brief presentation of the WLTP procedure compared to NEDC.

Comment: Chapter 3 discusses potential fuel savings.

Response: Thank you for your comment!

In this section, we focused on the analysis of a very simple model useful for fuel consumption evaluation.

Comment: The reference model was introduced in Chapter 4 but without the details that are necessary to recognize the overall complexity of the model. This chapter should be extended. The results of the fuel consumption analysis are presented in chapter 5.

Response: Thank you for your comment!

Following your observation, Section 4 was detailed in this revised version of the manuscript. Some information related to the construction of the simulation model is presented (lines 196-, lines 302-)

Comment: The authors proposed a comparison based on single values ​​of Qa and Qs (Qa and Qa should be described in the text as analytical and simulated values, respectively).

Response: Thank you for your observation!

The manuscript was revised (lines 263-):

Comment: Simple discussion is presented in chapter 6 (wrongly numbered as 5-th). I disagree with statement that results obtained for rolling resistance shows marginal differences. One can see that for speed exceeding 80 km/h  the differences of rolling resistance can be about 50 kN (c.a. 25%). At the end of the chapter simple application of the model is proposed.

Response: Thank you for your comment and careful reading of the paper!

The number of sections is updated in this section. Unfortunately, we mislabeled the y-axis of Figure  15 (in the present version). Furthermore, additional details are added in this revision. The details address the rolling resistance and the inertia force (lines 336-).

Comment: In my opinion, the authors should rewrite the manuscript. It will be valuable to show the accuracy of the model for a real traffic scenario (i.e. non-zero road gradients) compared to real measurement data.

Response: Thank you for your comment!

We hope that the current version of the manuscript presents the required information. To prove the efficiency of the analytical model, results from other driving cycles were added in Appendix 1.

Comment: 1. What is the main question addressed by the research? The work deals with the issue of estimating fuel consumption for given conditions of vehicle motion.

Response: Thank you for your comment!

The model was initially constructed as a simple tool for preliminary fuel consumption analysis. During the documentation, we found that although the analytical formulas are mentioned, the focus is mainly on very detailed models. With some work, we found that even the simplest models can provide good results. This is what we are showing in this paper.

Comment: 2. Do you consider the topic original or relevant in the field? Does it address a specific gap in the field? The problem is well known, as is the main method used by the authors (see e.g. https://doi.org/10.1016/j.trd.2011.05.008; https://doi.org/10.3390/en14237979. The model is one of the simplest models even among the black box models. Unfortunately, I do not identify the knowledge gap filled by the results of this paper

Response: Thank you for your comment!

The analytical model is well-known. This work only tries to point on the potential application of this simple model as a preliminary step for detailed models. We hope that the additional content of the revised manuscript will add some useful information for an interested reader.

Comment: 3. What does it add to the subject area compared with other published material? A comparison of the results obtained using the simple model and the MATLAB model is given. This is the most interesting part of the article.

Response: Thank you for your comment!

The information related to the use of MATLAB\Simulink model is detailed in this version.

Comment: 4. What specific improvements should the authors consider regarding the methodology? What further controls should be considered? The necessary list of improvements is as follows: -comparison of the result obtained from the model with the actual fuel consumption rate in the WLTP cycle for the vehicle under consideration -detailed description of the MATLAB model to recognize the overall complexity of the model and its shortcomings, -more extensive comparison of the analytical model and MATLAB in terms of the calculated force on the wheel and the use of appropriate measures to further discuss the differences -the accuracy of the backward model for a real traffic scenario (i.e. non-zero road gradients) compared to real measurement data -an extensive new part in which the authors present examples of the possible practical application of the model.

Response: Thank you for your comment!

The revised version of the manuscript presents results obtained for different values of the parameters defined in the analytical model. Some details were presented at the beginning of the document.

Unfortunately, read-driving datasets are available for analysis. However, we intend to include this analysis in the future works.

Comment: 5. Are the conclusions consistent with the evidence and arguments presented and do they address the main question posed? Conclusions should be rewritten to summarize the pros and cons of the backward model.

Response: Thank you for your comment! The Conclusion was revised.

Comment: 6. Are the references appropriate? Should be review (discussion about NEDC vs WLTP cycle is off topic) and completed with references to Vehicle Power Specific models.

Response: Thank you for your comment!

The reference list was checked. We consider the discussion related to WLTP and NEDC necessary as it details some aspect related to the benefits and shortcomings of the driving cycles. WLTP was especially designed to close the gap between benchmark results and real-driving conditions.

The Power-Based Model is now referenced in this revision, and we thank you for this suggestion. We think that it can be included in further developments of the current application (lines 296-299).

Comment: 7. Please include any additional comments on the tables and figures. Some explanations (see line 155) and equations (13, 14) are trivial. The same is about Fig. 1. Fig. 7 cannot be treated as a sufficient description of the MATLAB model. Figures 8, 9 and 10-13 should be redrawn to better show the outputs of both models. Figure 13 has an incorrect caption.

Response: Thank you for your comment!

The manuscript was revised, and the abovementioned equations were removed. Figures were reworked for an improved representation of the results.

Thank you!

Stefan TABACU

Dragos POPA

Reviewer 2 Report

1.For the rolling resistance coefficient in Eq. (3), is it proposed by authors? If not, please cite the reference. The same problem happens on Eq. (8), too. Are these parameters arisen from the experiments?

2. Fig. (4) and Fig (5) show the instantaneous energy at each time-step. However, in previous description, the energy is consumed during the time interval. These two figures may lead to a little misunderstanding. Moreover, the driving cycle should be pointed out.

3. There are some typing and format errors:(1) In table 1, the acronym of Brake-specific fuel consumption should be BSFC. (2) “Results presented in Figure 7 are ……” in line 228 should be “Figure 8”. (3) In line 243, there exist two blank spaces between the symbol of “$Q_{i/100}$” and “is”. And the symbol of “$Q_{i/100}$” should be “$Q_{i/100km}$” (4) In line 260-261, the sentence of “However, the battery capacity it was limited to 5.2 Ah to agree with the above mentioned hypothesis.” has grammar error.

4. Generally, the brake-specific fuel consumption isn’t a constant value in the range of the limited engine torque and speed. Why is the value of it in Table 1 constant?

5. In Fig. (8), it is suggested to provide the tracking error in value to analyze the accuracy of proposed model.

6. It should be added more detailed description for the phenomena appearing in Fig. (9).

7. The “Magic Formula” determined by Pacejka should be added the reference.

8. The value of identification by backward-facing analysis is different from it in Fig. (14). Please check the curve and the reference data.

Author Response

Dear Reviewer,

We kindly thank you for your careful reading of our submission. The observations and comments mentioned in the reviewing report are an important resource for the authors to improve the quality of the work.

We are also grateful for the opportunity of submitting a revised version.

Please receive the updated manuscript following the comments.

Thank you!

Best wishes!

Comment: 1.For the rolling resistance coefficient in Eq. (3), is it proposed by authors? If not, please cite the reference. The same problem happens on Eq. (8), too. Are these parameters arisen from the experiments?

Response: Thank you for your comment!

We added the required reference. We would like to mention that the source was previously mentioned in the references listed (Wong, J., Theory of ground vehicle); however we, unfortunately, missed to add this entry whenever it as used.

Thank you for considering this response!

Comment: 2. Fig. (4) and Fig (5) show the instantaneous energy at each time-step. However, in previous description, the energy is consumed during the time interval. These two figures may lead to a little misunderstanding. Moreover, the driving cycle should be pointed out.

Response: Thank you for your comment!

The term “time-step” is now replaced with “reference time interval”. Some details are added in the manuscript (lines 183-):

Thank you for considering this response!

Comment: 3. There are some typing and format errors:(1) In table 1, the acronym of Brake-specific fuel consumption should be BSFC. (2) “Results presented in Figure 7 are ……” in line 228 should be “Figure 8”. (3) In line 243, there exist two blank spaces between the symbol of “$Q_{i/100}$” and “is”. And the symbol of “$Q_{i/100}$” should be “$Q_{i/100km}$” (4) In line 260-261, the sentence of “However, the battery capacity it was limited to 5.2 Ah to agree with the above mentioned hypothesis.” has grammar error.

Response: Thank you for your comments and careful reading!

3.1. The text “BSFC” was updated in Table 1.

3.2. Text is updated in this revised version (line XX): “Results presented in Figure 8...”.

3.3. All the symbols, units and numbers indicated in the paper are inserted using the Equation Editor. Thus, it is possible to display blank spaces larger than the typical space.

3.4. The text was updated (line XX):

“The battery capacity it was limited to 5.2 Ah to agree with the abovementioned hypothesis.”

Thank you for considering this response!

Comment: 4. Generally, the brake-specific fuel consumption isn’t a constant value in the range of the limited engine torque and speed. Why is the value of it in Table 1 constant?

Response: Thank you for your comments!

We added some explanation in the revised version (lines 231-):

Comment: 5. In Fig. (8), it is suggested to provide the tracking error in value to analyze the accuracy of proposed model.

Response:  Thank you for your comment!

Following this observation, we added some information and updated the graphical representations (lines 372-).

Comment: 6. It should be added more detailed description for the phenomena appearing in Fig. (9).

Response: Thank you for your comment!

The revised manuscript presents some updated graphical representations (Figure 11 and Figure 12), as well the figures mentioned above.

Comment: 7. The “Magic Formula” determined by Pacejka should be added the reference.

Response: Thank you for your comment!

We apologize for this omission. The work of Hans Pacejka is a valuable resource for the field of vehicle dynamics. The text was updated for this version (line 324-)

Comment: 8. The value of identification by backward-facing analysis is different from it in Fig. (14). Please check the curve and the reference data.

Response: Thank you for your observation!

We check all the figures. The use of different reference speeds for the estimate of inertia sourced the difference in the previous version. Some explanations were provided.

Thank you!

Stefan TABACU

Dragos POPA

Reviewer 3 Report

This paper presents a methodology for the estimation of fuel consumption using backward-facing analysis. However, I have some remarks.

1. Some formulas are obvious and do not need to be listed, for example Eq.(13), Eq.(14)

2. The Introduction should be rewritten. The current research progress, and more contents should be provided. The references such as DOI: 10.1109/TTE.2022.3181201 and 10.1109/TMECH.2022.3156150, may be useful.

3. The lines in Figure 8 are indistinguishable. Different line types should be used. All figures in this paper should be double-checked.

4. The fuel consumption is closely associated with the performance of energy management strategy for hybrid electric vehicles. This part should be presented in this paper.

5. More validation results need to be provided.

Author Response

Dear Reviewer,

We kindly thank you for your careful reading of our submission. The observations and comments mentioned in the reviewing report are an important resource for the authors to improve the quality of the work.

We are also grateful for the opportunity of submitting a revised version.

Please receive the updated manuscript following the comments.

Thank you!

Best wishes!

This paper presents a methodology for the estimation of fuel consumption using backward-facing analysis. However, I have some remarks.

Comment: 1. Some formulas are obvious and do not need to be listed, for example Eq.(13), Eq.(14)

Response: Thank you for your comment!

The manuscript was revised, and the abovementioned equations were removed.

Comment: 2. The Introduction should be rewritten. The current research progress, and more contents should be provided. The references such as DOI: 10.1109/TTE.2022.3181201 and 10.1109/TMECH.2022.3156150, may be useful.

Response: Thank you for your observation!

We revised the introduction. Although we did not rewrite this section, additional information was added through the text to complete the discussion.

Thank you for considering this response!

Comment: 3. The lines in Figure 8 are indistinguishable. Different line types should be used. All figures in this paper should be double-checked.

Response: Thank you for your comment!

The revised manuscript presented some updated graphical representation and was completed to add some detailed information for the existing data (e.g., lines 372-).

.

Comment: 4. The fuel consumption is closely associated with the performance of energy management strategy for hybrid electric vehicles. This part should be presented in this paper.

Response:  Thank you for your observation!

We are looking at the potential of a hybrid vehicle; however, at this point, we do not have the knowledge to update the logic of the controller integrated into the electric energy management system.

Thank you for considering this response!

Comment: 5. More validation results need to be provided.

Response: Thank you for your comment!

We hope that the current version of the manuscript presents the required information. To prove the efficiency of the analytical model, results from other driving cycles were added in Appendix 1.

Thank you!

Stefan TABACU

Dragos POPA

Round 2

Reviewer 1 Report

Dear Authors, thank you for detailed explanations. Please check the numbers of equations.

Author Response

Dear Reviewer,

We thank you for your valuable contribution to our submission. We consider that your comments and observations are very important and provide the opportunity of improving our work to clearly and completely present our small contribution to this field.

Best wishes!

Stefan TABACU

Dragos POPA

Comment: Dear Authors, thank you for detailed explanations.

Response: Thank you!

Comment: Please check the numbers of equations.

Response: Thank you for the careful reading! The numbers of the equations are now updated.

Reviewer 2 Report

1. For the equation, there exist some question: (1) In Eq. (1), the symbol $\gamma$ and the description of symbol $\delta$ don’t match. Please check it. (2) In Eq. (12)~(14), what is the meaning of the symbol “\Delta d_{i,i-1}”

2. The figures in this paper seem to be not clear, please change the origin picture file.

3. The values of the empirical formula are mostly based on the test data of the corresponding reference. But the authors use it to compare with the Matlab/Simulink model. Are they matching?

4. The table between line 373-374 lacks of the caption. Please add it.

5. For the Fig. (11) and Fig. (12), the trend of the two curves should be described in more detail to analyze the difference between the two model.

Author Response

Dear Reviewer,

We thank you for your valuable contribution to our submission. We consider that your comments and observations are very important and provide the opportunity of improving our work to clearly and completely present our small contribution to this field.

Best wishes!

Stefan TABACU

Dragos POPA

Comment 1: For the equation, there exist some question: (1) In Eq. (1), the symbol $\gamma$ and the description of symbol $\delta$ don’t match. Please check it. (2) In Eq. (12)~(14), what is the meaning of the symbol “\Delta d_{i,i-1}”

Response: Thank you for the careful reading!

The symbol for the coefficient of the inertia force $\gamma$ is now updated.

An explanation is provided for “\Delta d_{i,i-1}” (line 194-)

„The distance travelled during the reference time interval from time  to time  is denoted by .” 

Comment 2: The figures in this paper seem to be not clear, please change the origin picture file.

Response: Thank you!

The figures are inserted from MATLAB using the clipboard mechanism of Windows. We tried different sizes, but we also looked at the dimensions. Furthermore, we would like to mention that the paper provides all the necessary details to construct these figures using MATLAB or a spreadsheet application.

Thank you for considering this response!

Comment 3: The values of the empirical formula are mostly based on the test data of the corresponding reference. But the authors use it to compare with the Matlab/Simulink model. Are they matching?

Response: Thank you for your comment! The backward-facing model uses empirical equations and describes a generic vehicle. The MATLAB/Simulink model represents a particular solution. Thus it is possible to use some specific details. However, the discussion related to the wheel force components gives the opportunity to evaluate each solution.

Thank you for considering this response!

Comment 4. The table between line 373-374 lacks of the caption. Please add it.

Response: Thank you for the careful reading! A caption is added for Table 2.

Table 2. Parametric analysis of ”

Comment 5: For the Fig. (11) and Fig. (12), the trend of the two curves should be described in more detail to analyze the difference between the two model.

Response: Thank you for your comment!

We updated the caption of Figure 12.

„The electrical component of the propulsion system results in a decrease in fuel consumption, which can be observed at the beginning of the driving cycle when the battery is charged at the nominal capacity.”

Thank you for considering this response!

Reviewer 3 Report

Thanks for your revision, and I have no comments.

Author Response

Dear Reviewer,

We thank you for your valuable contribution to our submission. We consider that your comments and observations are very important and provide the opportunity of improving our work to clearly and completely present our small contribution to this field.

Best wishes!

Stefan TABACU

Dragos POPA

Comment: Thanks for your revision, and I have no comments.

Response: Thank you!
